# The Conventional Weil Osteotomy Does Not Require Screw Fixation

**DOI:** 10.3390/jcm12020428

**Published:** 2023-01-05

**Authors:** Anastasia Boss, Eva Herrmann, Yves Gramlich, Alexander Klug, Oliver Neun, Sebastian Manegold, Reinhard Hoffmann, Sebastian Fischer

**Affiliations:** 1Department of Foot and Ankle Surgery, Berufsgenossenschaftliche Unfallklinik Frankfurt am Main, 60389 Frankfurt, Germany; 2Institute of Biostatistics and Mathematical Modelling, Goethe-Universität Frankfurt am Main, Theodor-Stern-Kai 7, 60596 Frankfurt am Main, Germany; 3Department for Trauma and Orthopaedic Surgery, Berufsgenossenschaftliche Unfallklinik Frankfurt am Main, 60389 Frankfurt, Germany

**Keywords:** Weil osteotomy, wedge-cut, flat-cut, metatarsalgia, claw toe, screw fixation

## Abstract

The Weil osteotomy is an established procedure to reduce plantar pressure in chronic metatarsalgia. Historically, the refixation of the displaced metatarsal head is performed by screw fixation. We aimed to demonstrate that screw fixation is not always necessary. Between 2016 and 2021, 155 patients with 278 Weil osteotomies (20 males and 135 females, mean age: 63 years) were retrospectively enrolled. Group A (*n* = 96) underwent 195 Weil osteotomies with screw fixation; group B (*n* = 59), 83 without screw fixation. Demographic, Visual Analog Scale Foot and Ankle (VAS-FA), SF-12 questionnaire, and toe mobility data were recorded. The mean follow-up period was 4.5 years. The mean VAS-FA was 75.5; mean SF-12 physical component summary, 42.0; and mean SF-12 mental component summary, 51.0. The overall revision rate was 20% (group A: 25%, group B: 10.2%), primarily for arthrolysis of the metatarsophalangeal joint in group A. Clinical comparisons showed no significant difference between the groups (*p* > 0.05). The revision rate was significantly higher in group A (*p* < 0.05), with equal satisfaction in clinical outcomes. Based on the available data, the need for regular screw fixation after a Weil osteotomy cannot be justified.

## 1. Introduction

The Weil osteotomy is an established procedure to reduce plantar pressure in chronic metatarsalgia. An inhomogeneous metatarsal parabola is usually present, with the second metatarsus considered to be too long. Increasingly, the Weil osteotomy has been performed to correct flexible claw toes, especially if associated with a subluxation or dislocation of the metatarsophalangeal joint. Historically, the refixation of the displaced metatarsal head is performed by screw fixation (Figure 1a,b).

Studies of smaller case series have already shown that, at least with a minimal incisional surgery (MIS) or percutaneous distal metatarsal mini-invasive osteotomy (DMMO), a screw fixation is not absolutely necessary [1,2,3,4]. All known procedures have postoperative limited mobility in the metatarsophalangeal joint, up to the so-called “floating toe,” in common [5]. This significantly reduces patient satisfaction with the otherwise safe procedure to reduce metatarsalgia.

We aimed to compare conventional Weil osteotomies with and without screw fixation, to determine the rate of delayed healing or arthrofibrosis in the metatarsophalangeal joint. We hypothesized that, due to the absence of prominent screw tips, there would be a lower rate of revision surgery.

## 2. Patients and Methods

### 2.1. Population

Between 2016 and 2021, 155 patients with 278 osteotomies (males, 20 (12.66%); females, 135 (85.44%); mean age: 63 years [range: 26–88 years]) were retrospectively enrolled in this comparative study. The demographic characteristics of both groups were equally distributed (Table 1). All patients were seen during foot surgery consultation at the study center (Figure 2). The diagnoses of metatarsalgia and claw toes were made on the basis of clinical examination and obligatory weight-bearing radiographs. All patients underwent isolated bone realignment without plantar plate repair: group A included 195 Weil osteotomies with screw fixation; group B, 83 osteotomies without screw fixation.

The mean follow-up duration for clinical outcomes was 4.5 years (range: 10–81 months). All procedures were performed in accordance with the 1964 Helsinki Declaration and its later amendments. The ethics committee of the institutional review board approved this study (2022-2813-evBO).

### 2.2. Inclusion and Exclusion Criteria

Patients with a minimum age of 18 years were included. There was no maximum age limit. Written informed consent was required prior to participation. The indication was based on underlying painful transfer metatarsalgia or claw toe of the lesser toes. Only surgeries performed at the study center were included.

Patients with complaints due to aseptic bone necrosis, underlying rheumatoid disease, and traumatic dislocation of the lesser metatarsophalangeal joints were excluded. Weil osteotomies as a revision procedure, and patients on chronic pain therapy were also excluded.

### 2.3. Surgical Procedure Using the Second Metatarsal as an Example

The patient was placed in the supine position. The subsequent procedure was performed under local or general anesthesia and an ankle tourniquet was obligatorily applied.

A dorsal incision of approximately 2.5–3.5 cm was made between the second and third metatarsals, entering between the short and long extensor digitorum tendon. Once the joint was fully exposed, a controlled shortening osteotomy was performed using a saw, as parallel as possible to the weight-bearing surface of the involved metatarsal: the end-to-end cut was started 1–2 mm below the dorsal border of the articular cartilage. Under manual pressure, the metatarsal head was displaced proximally.

In accordance with surgeon choice, all Weil osteotomies without screw fixation were performed with a wedge-cut technique (Figure 3 and Figure 4a,b).

In group B, the surgeons decided between wedge-cut and flat-cut techniques, depending on the length of the metatarsal. A total of three equally experienced surgeons were involved in the study. Two surgeons treated all patients in group A, another only the patients in group B. A simultaneous plantar plate repair was not performed in any patient, regardless of group.

### 2.4. Rehabilitation Protocol

After wound healing, usually two weeks postoperatively, pain-adapted full weight bearing was possible. A bandage or forefoot offloading shoe was worn for six weeks. If necessary, depending on the additional surgery of the first metatarsal, a longer non-weight-bearing period was observed. The lesser toes were allowed to be actively moved immediately after surgery.

### 2.5. Assessment Methods

The VAS-FA and the SF-12 questionnaires with physical and mental component summaries were collected to assess clinical outcome. Demographic data including BMI, preexisting conditions such as diabetes mellitus, and arterial hypertension (metabolic syndrome-associated), rheumatism, and nicotine abuse were obtained for each patient. In addition, postoperative toe mobility was recorded, with the decisive factor being the ability to establish toe ground contact. To this end, four grades were defined: unrestricted mobility; restricted mobility but able to touch the ground; incomplete “floating toe”; complete “floating toe”, not able to touch the ground.

In addition, a comparison of preoperative and postoperative radiographs was performed. Of particular interest was restoring the harmonious parabola of the forefoot and timely bone healing six weeks after surgery. Measurements were taken by an independent radiologist and two different surgeons.

### 2.6. Statistical Analysis

The primary objective was to demonstrate non-inferiority of the Weil osteotomy with screw versus without screw fixation at a mean follow-up of 4.5 years. When the present study was planned, studies investigating a similar question had a significantly smaller number of patients, which illustrates the power of the included data [1,2,3,6]. Statistical analyses were performed using SPSS v. 23 software (IBM Dtl. GmbH, Ehningen, Germany).

Furthermore, descriptive and explorative statistical analyses for the queried scores and evaluations of the pre- and postoperative radiographs (including within-group means, medians, minima and maxima, and standard deviations) were applied. Student’s *t*-test and ANOVA were used. The power of the study was 0.8, and significance level was set to *p* < 0.05, with a 95% confidence interval.

## 3. Results

In more than 70% of all cases, a hallux valgus deformity was corrected in addition to the Weil osteotomy. After a mean follow-up period of 4.5 years, the following clinical data were collected.

The mean VAS-FA was 75.76 (group A: 77.35, group B: 73.76); mean SF-12 physical component summary was 42.01 (group A: 41.55, group B: 42.80); and mean SF-12 mental component summary was 50.98 (group A: 49.97, group B: 52.67). Toe mobility was reported as not relevantly restricted in 72.66% of all osteotomies. After 22 osteotomies, complete or incomplete “floating toes” remained (group A: 17, group B: 5). Around 20% of all patients complained of a persistent conflict with the footwear. About half of all patients adapted the footwear with an orthotic insole or even modified the footwear itself, and 66% of both groups were satisfied with the procedure and would have it again (Table 2 and Table 3). No significant differences were observed between groups (*p* > 0.05). All patients reported strict adherence to the prescribed rehabilitation protocol, so there were no significant differences between the groups. The clinical results of the two techniques (wedge-cut or flat-cut technique) did not differ considerably; hence, a separate presentation of the results was omitted.

### Complications

The overall revision rate was 20%, with a significantly higher proportion in the screw fixation group (all *n* = 31; group A: *n* =26 (27.10%), group B: *n* =5 (8.47%); *p* = 0.005). The most common procedure was arthrolysis of the metatarsophalangeal joint, in group A with implant removal.

In addition, minor complications such as delayed wound healing, swelling, discomfort, and toe cramps were recorded. In three group A patients, five infections occurred with the need for premature implant removal and debridement. The clinical outcome of these patients did not differ significantly.

## 4. Discussion

The most important result of our study was that the two surgical options presented for the treatment of painful transfer metatarsalgia, which is based, among other things, on a “too long” metatarsal bone, led to predominantly good clinical results with a harmonious metatarsal parabola [7]. Above all, however, all results showed no significant difference in homogeneously distributed patient characteristics in both groups. Furthermore, the results are comparable with the current literature [8,9], both with the conventional Weil osteotomy with screw fixation, and with the increasingly propagated DMMO without screw fixation [1,3].

It is well known that inadequate correction with insufficient shortening or elevation of the affected metatarsal leads to recurrent metatarsalgia [10]. These recurrent complaints were observed in only 16 out of 278 metatarsals in the present study. Overcorrection, on the other hand, can result in transfer lesions to the adjacent metatarsals [9]. In the group with screw fixation, eleven new Weil osteotomies were performed; the underlying indication was almost equally divided between the problems described above. In the group without screw fixation, only one repeat Weil osteotomy was performed (*p* < 0.05).

All the surgical options described above, from the conventional Weil osteotomy using the wedge-cut or flat-cut technique, to the modified MIS procedures and DMMO, have the common problem of stiffness of the affected MTP joint. The increasingly performed minimally invasive procedures were designed to reduce this problem. The evidence in the direct comparison of Weil osteotomy and DMMO is sparse and inadequate; furthermore, an unsatisfactory incidence of floating toes of sometimes 30% or more remains with both procedures [11,12,13,14,15,16].

The “floating toe” rate of over 40% after Weil osteotomy reported by Garcı’a-Fernandez et al. was fortunately not observed in the present study’s patients, but is nevertheless a serious problem that is intrinsic to this procedure [17,18,19,20]. The “floating toe” rate in the present study was approximately 8% (toe mobility grades 3 and 4). However, it should be noted that a significant proportion of all patients had already received arthrolysis of the affected joint at the time of the follow-up examination. In the screw fixation group, the complaint regarding the limitation of motion together with discomfort implicating hardware led to 53 revisions of 195 treated metatarsals with implant removal and arthrolysis corresponding to 31 revisions of 155 treated patients. Only 5 of 59 patients (8.5%) in the group without screw fixation after Weil osteotomy required revision with arthrolysis, so the proportion was significantly lower than in the comparison group. Other studies showed revision rates ranging from <10% to well over 30% [14,17,21]. The literature even reports worrying infection rates of over 20% [22]. Even if the revisions with implant removal performed in the present study led to a better result, the authors believe that a revision rate of 25% or more is unacceptable either way. The surgical procedure must be optimized urgently and the indication critically questioned.

Regarding delayed union, all the patients, including those without screw fixation, were equally encouraged to move the corrected toes freely immediately after surgery. Only the need for non-weight bearing depended upon the accompanying intervention. An influence on the clinical outcome of Weil osteotomy could not be deduced. As prescribed, the majority of all patients received additional hallux valgus correction. In the case of an arthrodesis, for example, the Lapidus arthrodesis with non-weight bearing of 4–6 weeks, this restriction had no influence on the result compared to the otherwise early full weight bearing (r = 0.053).

The need for screw fixation was previously based on the assumption that omission thereof would lead to delayed union, malunion, or non-union of the osteotomized metatarsals, although most studies do not actually show a high rate of osseous healing failure [17,19]. In the present study, delayed bone healing was documented in only five cases, remarkably, entirely in the screw fixation group. Non-union or even osteonecrosis of the affected metatarsal head was not observed in any patient.

Although Weil osteotomies have been an established procedure in foot surgery since 1985 at the latest, there is still disagreement about the exact execution of the technique. Neither the influence of the wedge-cut or flat-cut technique, nor the necessary extent of displacement of the metatarsal head, have been clearly proven. All osteotomies without screw fixation in the present study were performed using the wedge-cut technique. Thus, beyond the original question, it can be said that the removal of a bone wedge did not lead to an increased rate of failed healing in group B (Figure 4a,b). A prospective comparison of Weil osteotomy wedge-cut and flat-cut techniques is the subject of current research [23].

The omission of screw fixation also leads to an expected reduction in costs. This fact was not investigated in the present study but remains obvious. The fact that patients without screw fixation require significantly fewer revisions also suggests a reduction in the economic burden.

This study had some limitations. This was a monocentric study with a retrospective design, and clinical scores and toe mobility were not collected preoperatively. The indication for surgery was presumably influenced by surgeon experience and preference, as was the decision for or against refixation of the displaced metatarsal head. Particularly in the case of simultaneous hallux correction or extended foot surgery, it is difficult to present the scores for the Weil surgery alone. The available data do not allow a statement about the pressure distribution after osteotomy. Consequently, no distinction can be made between the wedge-cut and flat-cut techniques. However, both groups were equally subjected to this bias. Our results appear suitable for prospective comparison. In this study, all surgeons involved performed the procedures to be compared with equal frequency.

## 5. Conclusions

With equal satisfaction in clinical outcomes, the group with screw fixation after Weil osteotomy had a significantly higher revision rate. The need to wear adapted insoles or shoes was also equally frequent in all patients. Based on the available data, the need for regular screw fixation after a Weil osteotomy cannot be justified.

## Figures and Tables

**Figure 1 jcm-12-00428-f001:**
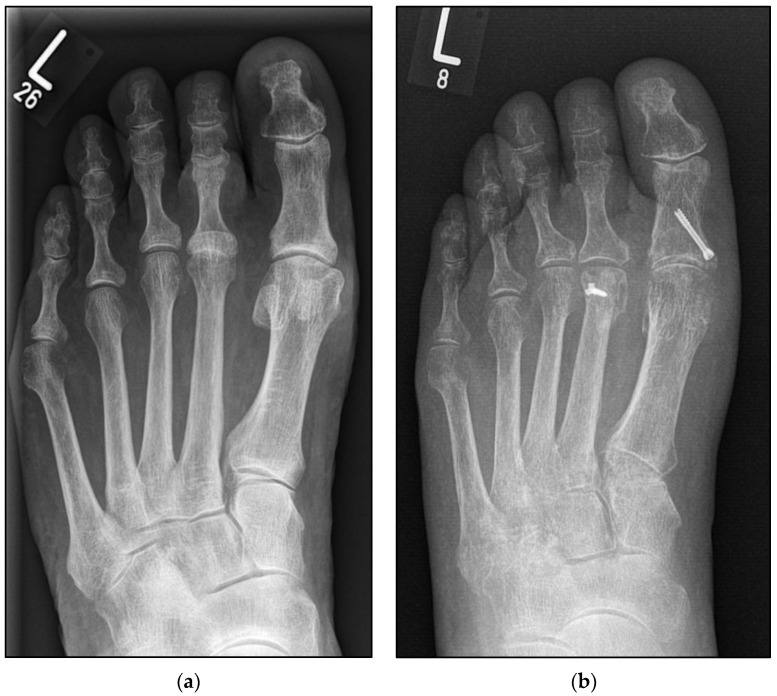
Pre- and postoperative radiographic findings of combined Weil osteotomy (group A with screw) and hallux valgus surgery, left foot. (**a**) Anteroposterior view preoperative, (**b**) anteroposterior view 3 months postoperative.

**Figure 2 jcm-12-00428-f002:**
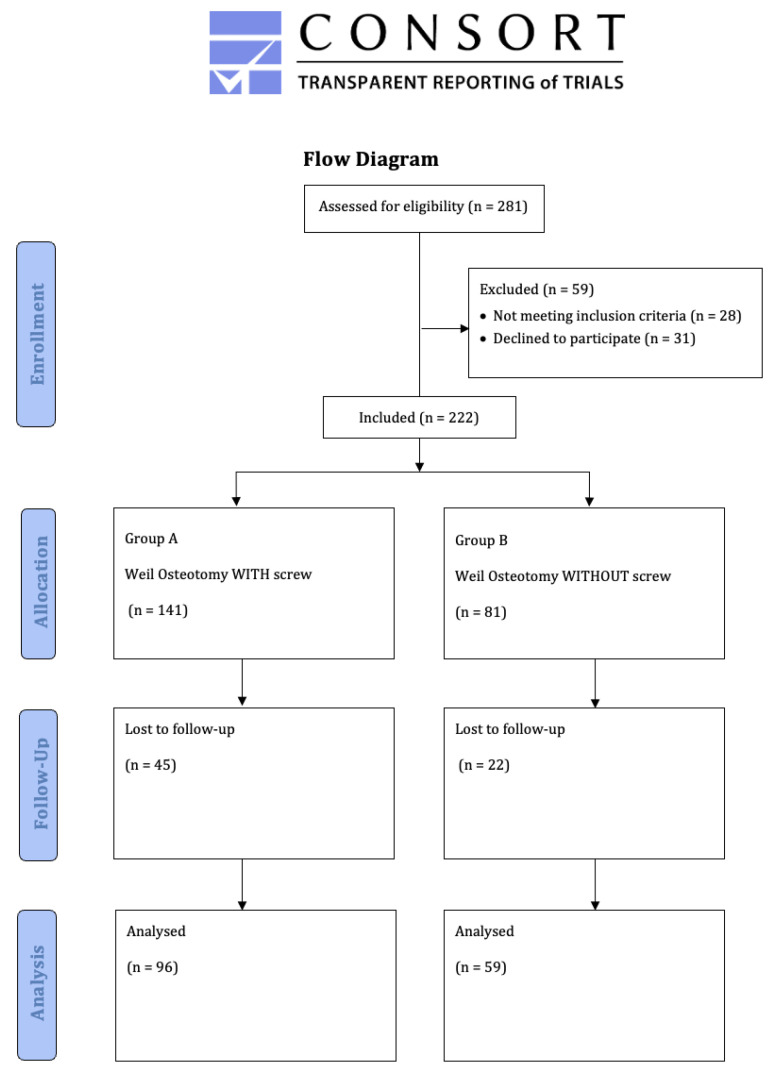
Study flow chart.

**Figure 3 jcm-12-00428-f003:**
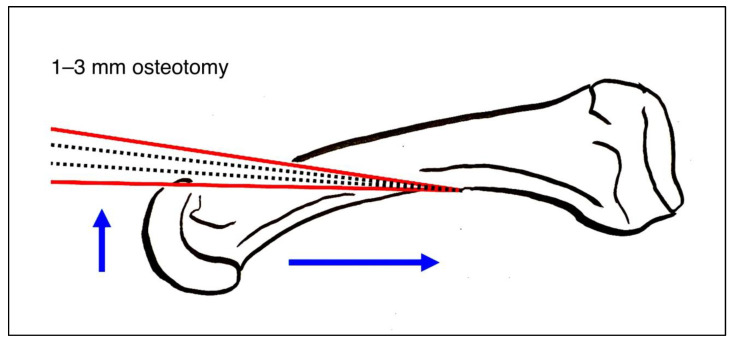
Weil osteotomy in wedge-cut technique.

**Figure 4 jcm-12-00428-f004:**
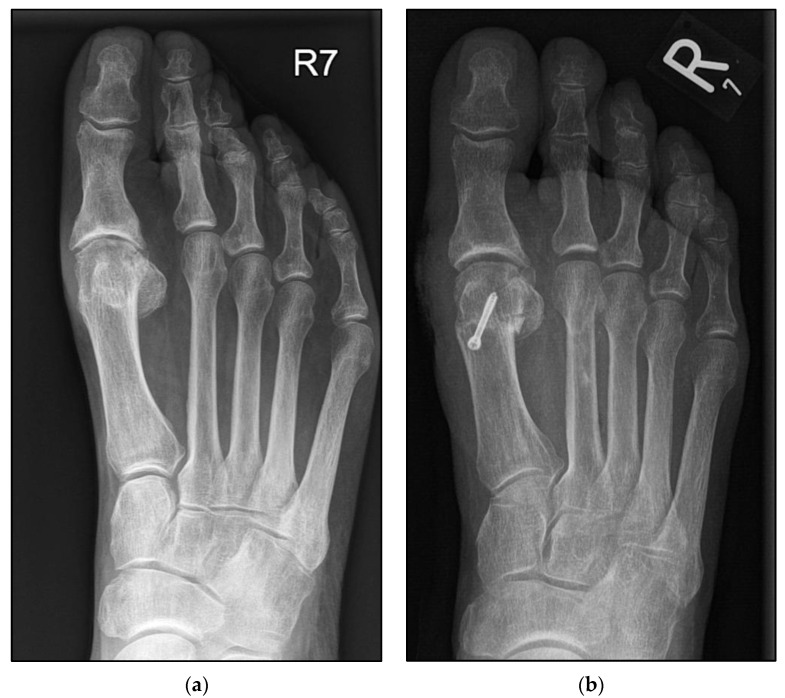
Pre- and postoperative radiographic findings of combined Weil osteotomy of metatarsal 2 (group B without screw) and hallux valgus surgery, left foot. (**a**) Anteroposterior view preoperative, (**b**) anteroposterior view 6 weeks postoperative.

**Table 1 jcm-12-00428-t001:** Patient characteristics.

Characteristic		With Screw (*n* = 96)	Without Screw (*n* = 59)	All (*n* = 155)	*p*
Age, years	Mean	63.37	62.37	62.99	0.620
	SEM	1.31	1.39	0.97	
	Minimum	26.00	33.00	26.00	
	Maximum	88.00	83.00	88.00	
BMI, kg/m^2^	Mean	26.91	25.72	26.465	0.174
	SEM	0.56	0.62	0.422	
	Minimum	17.78	19.300	17.78	
	Maximum	42.50	39.40	42.50	
Sex, *n* (%)	Male	17 (17.71)	3 (5.08)	20 (12.66)	0.023
	Female	79 (82.29)	56 (94.92)	135 (85.44)	
Affected side, *n* (%)	Left	46 (47.91)	34 (57.63)	80 (51.61)	0.048
	Right	40 (41.67)	25 (42.37)	65 (41.94)	
	Both sides	10 (10.42)	0 (0.00)	10 (6.45)	
Smoker, *n* (%)	Yes	13 (13.54)	6 (10.17)	19 (12.26)	0.537
	No	83 (86.46)	53 (89.83)	136 (87.74)	
Preexisting conditions (multiple answers), *n* (%)	Metabolic-syndrome-associated	39 (40.63)	24 (40.68)	63 (40.64)	0.726
	Rheumatism	6 (6.25)	2 (3.39)	8 (5.16)	
	Others	5 (5.21)	5 (8.47)	10 (6.45)	
	None	21 (21.88)	11 (18.64)	32 (20.65)	
	n.a.	25 (26.04)	17 (28.82)	42 (27.09)	

BMI, body mass index; SEM, standard error of the mean.

**Table 2 jcm-12-00428-t002:** Clinical outcomes according to treatment, based on the number of treated patients.

Measurements		With Screw (*n* = 96)	Without Screw (*n* = 59)	All (*n* = 155)	*p*
Follow-up in months	Mean	59.30	44.11	54.77	<0.001
	Range	(10.00–81.00)	(10.00–79.00)	10.00–81.00	
Number of affected toes (MT 2–5)		195	83	278	
Affected side	Left	46 (47.91)	34 (57.63)	80 (51.61)	0.048
	Right	40 (41.67)	25 (42.37)	65 (41.94)	
	Both sides	10 (10.42)	0 (0.00)	10 (6.45)	
Simultaneous hallux valgus surgery, *n* (%)	Yes	64 (66.67)	48 (81.36)	112 (72.26)	0.048
	No	32 (33.33)	11 (18.64)	43 (27.74)	
VAS-FA	Mean	77.35	73.76	75.46	0.343
	SEM	2.22	3.17	1.40	
	Minimum	12	19.84	12.85	
	Maximum	99.21	99.40	99.40	
SF-12 (physical component summary)	Mean	41.55	42.80	42.01	0.527
	SEM	1.21	1.57	0.96	
	Minimum	14.90	18.19	14.90	
	Maximum	60.36	63.53	63.53	
SF-12 (mental component summary)	Mean	49.97	52.67	50.98	0.134
	SEM	1.13	1.34	0.87	
	Minimum	26.56	22.98	22.98	
	Maximum	66.84	66.44	66.84	
Revision surgery needed, *n* (%)	Yes	26 (27.10)	5 (8.47)	31 (20.00)	0.005
	No	70 (72.90)	54 (91.53)	124 (80.00)	
Would you have the procedure again?	Yes	64 (66.67)	39 (66.10)	103 (66.45)	0.798
	No	23 (23.95)	16 (27.12)	39 (25.16)	
	Undecided	9 (9.34)	4 (6.8)	13 (8.39)	
Footwear (multiple answer), *n* (%)	Orthotic insoles	52 (54.17)	31 (52.54)	83 (53.55)	0.705
	Shoe adaption	6 (6.25)	2 (3.39)	8 (5.16)	
Conflict with the shoe, *n* (%)	Yes	27 (28.13)	9 (15.25)	36 (23.23)	0.075
	No	63 (65.63)	45 (76.27)	108 (69.68)	
	n.a.	6 (6.25)	5 (8.48)	11 (7.10)	

SEM, standard error of the mean; MT, metatarsal; SF-12, 12-Item Short Form Health Survey.

**Table 3 jcm-12-00428-t003:** Clinical outcome according to treatment, based on the number of treated metatarsals.

Measurements		With Screw (*n* = 195)	Without Screw (*n* = 83)	All (*n* = 278)	*p*
Affected toes	MT 2 and 3	37 (38.54)	22 (37.29)	59 (38.06)	<0.001 */ 0.877
	MT 2–4	21 (21.88)	1 (1.70)	22 (14.19)	
	MT 2–5	6 (6.25)	0 (0.00)	6 (3.87)	
	MT 3 and 4	1 (1.04)	0 (0.00)	1 (0.65)	
	MT 4 and 5	1 (1.04)	0 (0.00)	1 (0.65)	
	Isolated MT 2	24 (25.00)	32 (45.24)	56 (36.13)	
	Isolated MT 3	2 (2.08)	1 (1.70)	3 (1.94)	
	Isolated MT 4	1 (1.04)	3 (5.09)	4 (2.58)	
	Isolated MT 5	3 (3.13)	0 (0.00)	3 (1.94)	
	All over MT 2	88 (45.13)	54 (65.06)	142 (51.08)	
Toe mobility 1–4, *n* (%) **	1	148 (75.90)	54 (65.10)	202 (72.66)	0.396
	2	16 (8.21)	17 (20.48)	33 (11.87)	
	3	16 (8.21)	5 (6.02)	21 (7.55)	
	4	1 (0.51)	0 (0.00)	1 (0.36)	
	n.a.	14 (7.18)	7 (8.43)	21 (7.55)	
Minor complications (multiple answers), *n* (%)					
	Delayed wound healing	9 (4.62)	4 (4.82)	6 (2.16)	0.320
	Delayed union	5 (2.56)	0 (0.00)	5 (1.80)	
	Persistent swelling	3 (1.53)	2 (2.40)	5 (1.80)	
	Toe cramps	8 (4.10)	3 (3.61)	11 (3.96)	
	Dysesthesia	2 (1.03)	5 (6.02)	7 (2.52)	
	Recurrent metatarsalgia	13 (6.67)	3 (3.61)	16 (5.76)	
Revision surgery (multiple answers), *n* (%)					
	Implant removal and arthrolysis	53 (27.18)	-	53 (19.06)	0.015
	Isolated MTP joint arthrolysis	5 (2.56)	9 (10.84)	14 (5.04)	
	Revision Weil osteotomy	11 (5.64)	1 (1.20)	12 (4.32)	
	Infection debridement	5 (2.56)	0 (0.00)	5 (1.80)	

SEM, standard error of the mean; MT, metatarsal; MTP, metatarsophalangeal joint; * comparison regardless of group allocation, ** 1: unrestricted mobility, 2: restricted mobility but can touch the ground, 3: incomplete floating toe, 4: complete floating toe not able to touch the ground.

## Data Availability

All data intended for publication are included in the manuscript.

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
