# Peer review of "The Conventional Weil Osteotomy Does Not Require Screw Fixation"

_jcm, 2023, doi:10.3390/jcm12020428_

Round 1
Reviewer 1 Report
For the next study presure distribution analysis would have been interesting.
The article confirms that similar to minimal invasive distal metatarsal osteotomies also Weil osteotomy can be performed without screw fixation with a lower need for revision surgery.
Figure 4 - Pre- and postoperative radiographic findings does not show the x-rays of the same patient.
Please correct that.
Author Response
Dear Reviewer, thank you for the helpful comments. the following changes have been added:
1. "For the next study presure distribution analysis would have been interesting."
Reviewer 2 Report
One concern is that all Weil Osteotomies performed with screw fixation were performed by two surgeons and the osteotomies without screw fixation were performed by another surgeon. A more appropriate design would randomize with or without screw fixation across all patients and have all three surgeons perform the osteotomy with or without screw fixation. I suspect surgeon comfort is involved but this introduces a difference in surgical techniques that could explain the difference in revision rate as well. Nevertheless, the sample sizes are significant and the paper does appear to answer the primary question. A repeat study addressing this study design concern would answer the clinical question more effectively. Interestingly, fusion rate was not particularly impacted by osteotomy without screw fixation. The revision rate of 20% in the screw fixation group is high but within reported literature rates.
Author Response
Dear Reviewer, thank you for pointing out the difference in surgeon involvement.
This fact is indeed a relevant limitation and has been noted as such in the manuscript.
- "Our results appear suitable for prospective comparison. In this study, all surgeons involved should perform the procedures to be compared with equal frequency. "